# Ex vivo and in vivo CRISPR/Cas9 screenings identify the roles of protein N-glycosylation in regulating T-cell activation and functions

Yu Hong[1,2†], Xiaofang Si[1,2,3†], Wenjing Liu[1,2,3†], Xueying Mai[1,2,4†], Yu Zhang[1,2,3,4*]

[1]Chinese Institute for Cancer Research, Chinese Institutes for Medical Research, Beijing, China; [2]National Institute of Biological Sciences, Beijing, China; [3]Chinese Academy of Medical Sciences & Peking Union Medical College, Beijing, China; [4]School of Basic Medical Sciences, Capital Medical University, Beijing, China

## eLife Assessment

The **important** study uses genome-wide CRISPR/Cas9 screenings to identify a novel target B4GALTZ1 that is implicated in modulating CD8+ T cell function in the context of anti-tumor immunity. The strength of evidence is **solid** but could benefit from more detail, particularly to verify the efficiency of knockout in their single gene KO lines and identification of N-glycosylation sites of TCR and CD8s. This work highlights the role of protein N-glycosylation, particularly B4GALT1 deficiency, in regulating CD8 function and anti-tumor immunity.

*For correspondence:
zhangyu@cimrbj.ac.cn

†These authors contributed equally to this work.

**Abstract** Cytotoxic CD8[+] T-cells play central roles in tumor immunotherapy. Understanding the mechanisms that regulate development, differentiation, and functions of cytotoxic CD8[+] T-cells leads to the development of better immunotherapies. By combining primary T-cell culture and a syngeneic mouse tumor model with both genome-wide and custom CRISPR/Cas9 screenings, we systematically identified genes and pathways that regulate PD-1 expression and functions of CD8[+] T-cells. Among them, inactivation of a key enzyme in glycoconjugate biosynthesis, beta 1,4-galactosyltransferase 1 (B4GALT1), leads to significantly enhanced T-cell receptor (TCR) activation and functions of CD8[+] T-cell. Interestingly, suppression of B4GALT1 enhances functions of TCR-T-cells, but has no effect on chimeric antigen receptor T (CAR-T) cells. We systematically identified the substrates of B4GALT1 on CD8[+] T-cell surface by affinity purification and mass spectrometry analysis, which include protein components in both TCR and its co-receptor complexes. The galactosylation of TCR and CD8 leads to reduced interaction between TCR and CD8 that is essential for TCR activation. Artificially tethering TCR and CD8 by a TCR-CD8 fusion protein could bypass the regulation of B4GALT1 in CD8[+] T-cells. Finally, the expression levels of B4GALT1 normalized to tumor-infiltrated CD8[+] T-cells in tumor microenvironment are significant and negatively associated with prognosis of human patients. Our results reveal the important roles of protein N-glycosylation in regulating functions of CD8[+] T-cells and prove that B4GALT1 is a potential target for tumor immunotherapy.

## Introduction

Cytotoxic CD8[+] T-cells play a central role in tumor immunotherapy. The presence and activation status of CD8[+] T-cells in tumors are efficient biomarkers to predict prognostic and therapeutic efficacy for patients receiving immunotherapy (*Spencer et al., 2016*; *Havel et al., 2019*). Immune checkpoint

inhibitors (e.g., anti-PD-1, anti-PD-L1, and anti-CTLA4 antibodies) targeting the reactivation of tumor-infiltrated cytotoxic CD8[+] T-cells have shown amazing clinical benefits in treating various human tumors (*Hodi et al., 2003*; *Brahmer et al., 2012*; *Topalian et al., 2012*; *Ribas and Wolchok, 2018*; *Cercek et al., 2022*). Chimeric antigen receptor T (CAR-T) and T-cell receptor-engineered T (TCR-T) cells, which are exogenous cytotoxic CD8[+] T-cells that are genetically engineered to directly target cancer cells, have demonstrated promising clinical efficacy against some human tumors (*June et al., 2018*; *Rafiq et al., 2020*). Fully elucidating the mechanisms regulating development, differentiation, and functions of cytotoxic CD8[+] T-cells paves ways for the development of better tumor immunotherapies.

Expression of programmed cell death protein 1 (PD-1) can be induced in a wide variety of immune cell types (*Agata et al., 1996*; *Yu et al., 2016*; *Gordon et al., 2017*). For example, TCR activation induces the expression of PD-1 on T-cell surface. The interaction between the PD-1 receptor and its ligand, PD-L1, reduces TCR signals to suppress the immune system. When tumors evade attacks from the immune system, cancer cells upregulate their surface PD-L1 expression to prevent attack by endogenous cytotoxic CD8[+] T-cells. Antibodies blocking the interaction between PD-1 and PD-L1 have been proven to be effective in human tumor immunotherapy and show high efficacy in many tumor types (*Brahmer et al., 2012*; *Topalian et al., 2012*). Understanding how immune cells regulate their PD-1 expression is one of the most important aspects of tumor immunology research. Although a few regulators of PD-1 expression in cytotoxic CD8[+] T-cells have been identified recently (*Park et al., 2016*; *Stephen et al., 2017*; *Okada et al., 2017*; *Meng et al., 2018*; *Zhou et al., 2020*), an unbiased systematic screening is still missing.

Here, by combining ex vivo primary memory CD8[+] T-cell culture and an in vivo syngeneic mouse tumor model with genome-wide and custom CRISPR/Cas9 screenings, we systematically identified genes and pathways that regulate PD-1 expression and functions of CD8[+] T-cells. Among them, inactivation of a key enzyme in glycoconjugate biosynthesis, that is, B4GALT1, enhances PD-1 expression, TCR activation, and functions of mouse CD8[+] T-cells both in vitro and in vivo. Similar roles of B4GALT1 were also observed in human CD8[+] T-cells. Mechanistic studies indicate that B4GALT1 regulates CD8[+] T-cell function via a TCR-dependent pathway through cell surface protein glycosylation. Systematic LC-MS analysis further identified that proteins in the TCR-complex and its co-receptors are direct substrates of B4GALT1. Mechanistic studies indicate that B4GALT1-deficient T-cells showed stronger TCR-CD8 interaction and enhanced TCR activation than wild-type control, which can be bypassed by artificially tethering of TCR with CD8 when a CD8β-CD3ε fusion protein was overexpressed. These results support a model that B4GALT1 modulates T-cell functions by direct galactosylation of TCR and CD8, which prevents the interaction between them. Finally, the expression of B4GALT1 showed a significant negative correlation with prognosis of human cancer patients when normalized to tumor-infiltrated CD8[+] T-cells. Taken all together, the results reveal the important roles of protein N-glycosylation in regulating functions of CD8[+] T-cells and prove that B4GALT1 is a potential target for tumor immunotherapy.

## Results

### Ex vivo and in vivo CRISPR/Cas9 screenings identify genes and pathways that regulate PD-1 expression and functions of CD8[+] T-cells

We set up an ex vivo genome-wide CRISPR/Cas9 screening system to identify genes and pathways that regulate PD-1 expression in mouse primary CD8[+] T-cells (*Figure 1a*). In brief, splenic CD8[+] T-cells from Cas9-EGFP/OT-I mice were infected with a retroviral whole-genome guide RNA (gRNA) library following ovalbumin (Ova) peptide stimulation. After puromycin selection, the memory CD8[+] T-cells were re-stimulated by co-culturing with B16F10-OVA cells. The highest and lowest 5% PD-1[+] cells were isolated by fluorescence-activated cell sorting (FACS) and defined as PD-1[high] and PD-1[low] populations, respectively (*Figure 1—figure supplement 1a*). The distributions of individual gRNAs in the whole-genome library in those subpopulations, as well as in input cells, were revealed by next-generation sequencing. As shown in *Figure 1b*, *Pdcd1* and previously identified PD-1 regulators, such as *Satb1* (*Stephen et al., 2017*) and *Fut8* (*Okada et al., 2017*), were successfully identified as positive controls. The top candidates were verified by single sgRNA knockout (*Figure 1—figure supplement 2*), and most of them showed matched results as in screenings both at RNA and cell surface protein expression levels (*Figure 1c*). Gene set enrichment analysis (GSEA) identified several Kyoto Encyclopedia

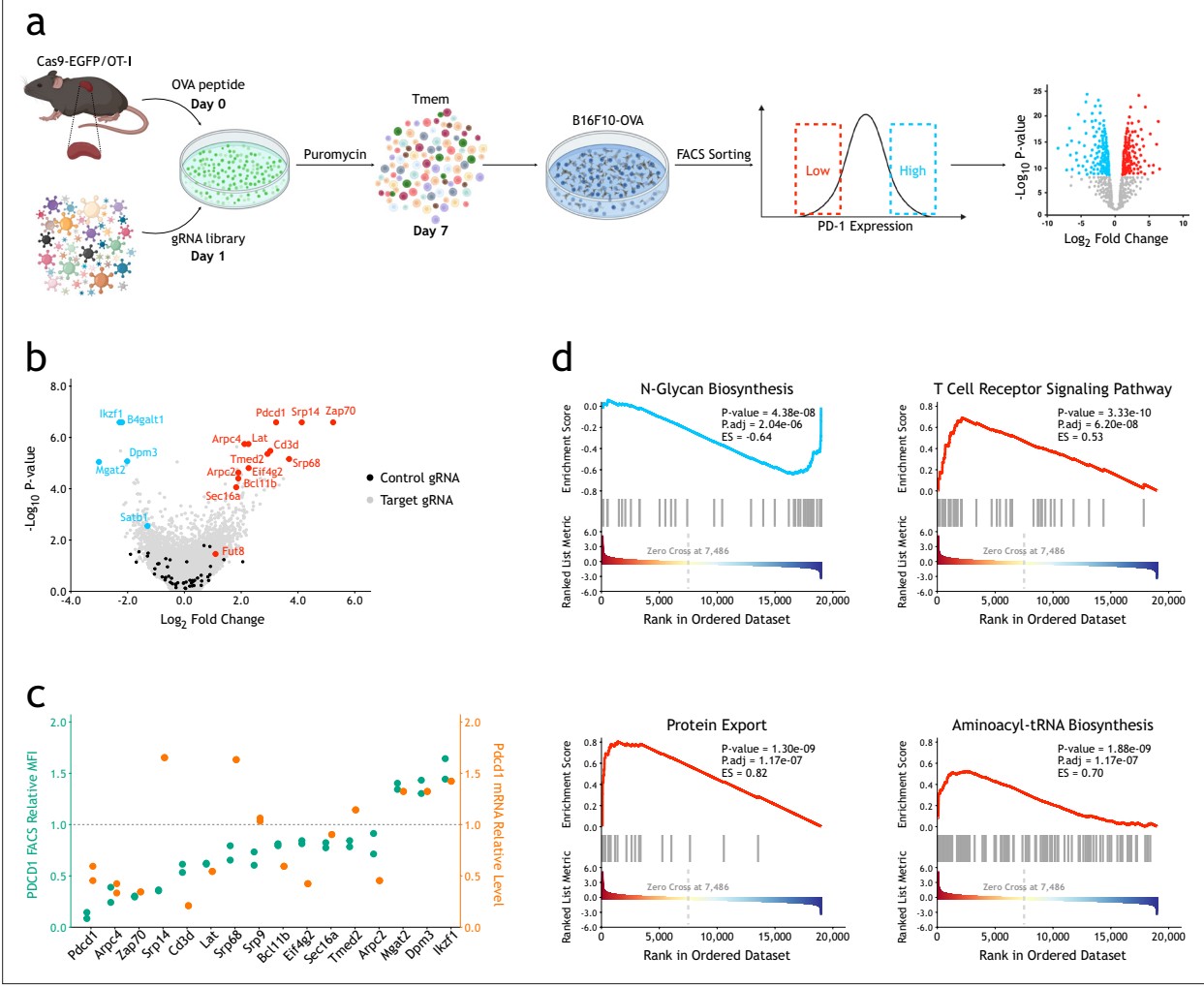

**Figure 1.** Ex vivo CRISPR/Cas9 screenings identify genes and pathways that regulate PD-1 expression of CD8[+] T-cells. (**a**) Schematic view of ex vivo CRISPR/Cas9 screening in mouse primary CD8[+] T-cells. (**b**) Volcano plot showing results of ex vivo CRISPR/Cas9 genome-wide screenings. The screenings were repeated independently once. The p-values were calculated using the α-robust rank aggregation (α-RRA) algorithm in MAGeCK. (**c**) Verification of candidate genes by individual single gRNAs. The relative expression levels of surface PD-1 protein and PD-1 mRNA were measured by FACS as mean fluorescent intensity (MFI) and RT-qPCR, respectively. The verification assays were biologically replicated twice. (**d**) GSEA of significantly enriched KEGG pathways in genome-wide screening. The enrichment score (ES) and statistical significance were calculated using the clusterProfiler (version 3.12.0) R package.

The online version of this article includes the following source data and figure supplement(s) for figure 1:

**Figure supplement 1.** Multiple components in N-glycan biosynthesis pathway were identified in genome-wide screenings for PD-1 regulators in CD8[+] T-cells.

**Figure supplement 2.** Gene knockout efficiency detected by T7E1 assay.

**Figure supplement 2—source data 1.** Original files for PAGE gel images for *Figure 1—figure supplement 2*.

**Figure supplement 2—source data 2.** PDF file containing original PAGE gel for *Figure 1—figure supplement 2*, indicating the genes, gRNAs, and relevant bands.

of Genes and Genomes (KEGG) pathways significantly involved in regulation of PD-1 expression in CD8[+] T-cells (*Figure 1d*). As expected, genes that are known to be involved in the T-cell receptor (TCR) activation pathway such as *Cd3d*, *Zap70*, and *Lat* were among the top ones identified. Interestingly, genes involved in protein export pathway (such as *Srp14*, *Srp68*, *Sec16A*) and in aminoacyl tRNA biosynthesis (such as *Mars*, *Hars2*, *Eprs*) were significantly increased in PD-1[low] populations. In addition, we also identified and verified genes required for N-glycan biosynthesis, including *B4galt1*,

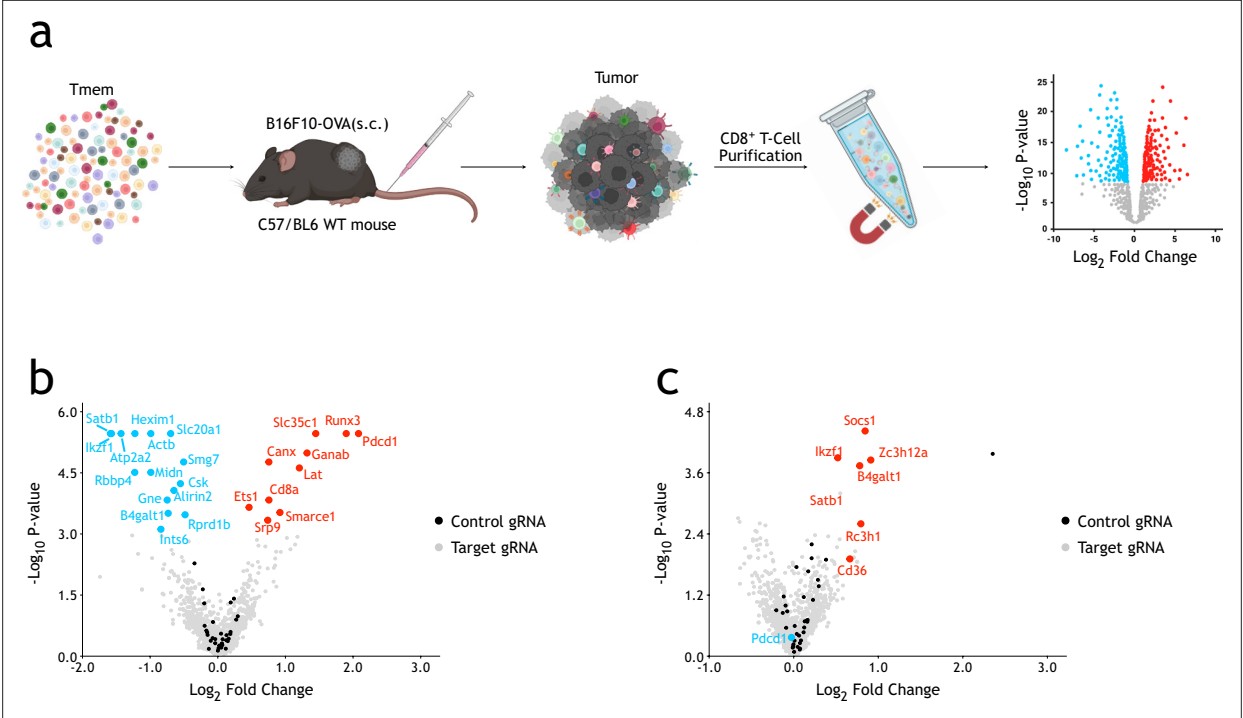

**Figure 2.** In vivo CRISPR/Cas9 screenings with a custom gRNA library identify genes that regulate functions of CD8+ T-cells in tumor microenvironment. (**a**) Schematic view of in vivo CRISPR/Cas9 screening in mouse primary CD8+ T-cells. (**b**) Volcano plot showing results of ex vivo CRISPR/Cas9 screening. The screenings were repeated independently once. The p-values were calculated using the α-RRA algorithm in MAGeCK. (**c**) Volcano plot showing results of in vivo CRISPR/Cas9 screenings. The screenings were repeated independently once. The p-values were calculated using the α-RRA algorithm in MAGeCK.

*Mgat2*, and *Dpm3*, which could negatively regulate the expression of PD-1 (*Figure 1—figure supplement 1b–d*).

To verify these candidate genes in a high-throughput manner, we synthesized a custom gRNA library containing 4617 gRNAs targeting the top candidate genes obtained from whole-genome screenings and 105 intergenic control gRNAs. The gRNA library-infected Cas9-EGFP/OT-I memory CD8+ T-cells were restimulated by co-culture and PD-1 regulators were enriched using the same strategy as whole-genome screening (*Figure 1a*). As shown in *Figure 2b*, most of the significant PD-1 regulators in the top list of genome-wide screening could also be enriched in the custom small library screening, which indicates the reliability of our ex vivo screenings. To test potential functions of these candidate genes in the context of tumor microenvironment, custom gRNA-library-infected Cas9-EGFP/OT-I memory CD8+ T-cells were transplanted into wild-type (WT) C57BL/6J mice inoculated subcutaneously (s.c.) with B16F10-OVA tumors to screen for genes regulating CD8+ T-cell functions in vivo. After 7 days, tumor-infiltrated CD8+ T-cells were collected for gRNA sequencing (*Figure 2a*). We successfully identified several positive control genes (*Wei et al., 2019*; *Zhao et al., 2021*; *Ma et al., 2021*; *Xu et al., 2021*) such as *Socs1*, *Regnase-1* (*Zc3h12a*), *Rc3h1*, and *Cd36*, indicating our in-vivo screening system worked robustly (*Figure 2c*). Interestingly, inactivating *B4galt1*, which encodes beta-1,4-galactosyltransferase 1, showed significant phenotypes in both ex vivo and in vivo screenings.

## B4GALT1 suppression in CD8+ T-cells activates TCR signaling and enhances T-cell functions both in vitro and in vivo

B4GALT1 is one of the seven beta-1,4-galactosyltransferases that transfer galactose in a beta 1–4 linkage to similar acceptor sugars, including N-acetylglucosamine (GlcNAc), glucose (Glc), and xylose (Xyl). More specifically, B4GALT1 uses UDP-galactose and N-acetylglucosamine for the production of galactose beta-1,4-N-acetylglucosamine (*Rodeheffer and Shur, 2002*). In addition to glycoconjugate biosynthesis, B4GALT1 can also form a heterodimer with a-lactalbumin (LALBA) as lactose synthetase in lactating tissues. Although B4GALT1 is expressed ubiquitously, its roles in regulating interaction and

adhesion of immune cells have been suggested (*Cheng et al., 2010*; *Han et al., 2010*; *Gómez-Henao et al., 2021*; *Cui et al., 2023*; *Hsu et al., 2024*; *De Vitis et al., 2019*).

Infection of Cas9-EGFP/OT-I CD8[+] T-cells with different single gRNAs targeting B4galt1 resulted in significantly increased surface PD-1 expression and PD-1 mRNA levels both before and after co-culture with B16F10-OVA cells (*Figure 3a* and *Figure 4—figure supplement 1d*). Such phenotypes could be rescued by overexpression of either a short or long isoform of mouse *B4galt1* cDNA (*Figure 3b*), suggesting that biosynthetic function, but not ligand-induced signal transduction, of B4GALT1 (*Rodeheffer and Shur, 2002*) is responsible for suppression of PD-1 expression. In addition, *B4galt1*-deficient OT-I T-cells showed enhanced expression of T-cell activation and cytotoxic markers such as IFNγ and TNFα (*Figure 3c*), as well as in vitro targeted cell killing activity when co-cultured with target cells, B16F10-OVA (*Figure 3d*). To verify whether B4GALT1 has similar functions in human T-cells, we infected primary human CD8[+] T-cells with NY-ESO-1 TCR constructs containing shRNA expression cassettes targeting human B4GALT1 (*Robbins et al., 2008*; *Figure 3e*). As shown in *Figure 3f*, compared with control, knockdown of human B4GALT1 enhanced targeted killing of human A375 cells in vitro. Increased secretion levels of IFNγ and TNFα have been observed after knockdown of B4GALT1 (*Figure 3g*). Surprisingly, knockout of B4GATL1 could not affect hCD19-CAR-mediated in vitro killing of Nalm6 cells (*Figure 3—figure supplement 1*). When stimulated with only anti-CD3/28 antibodies in the absence of target cells, B4GALT1 knockout OT-I cells also did not show significantly enhanced expression of IFNγ and TNFα, compared with controls (*Figure 3—figure supplement 2*). These results indicate that B4GALT1 may enhance T-cell functions via a TCR- and target-cell-dependent pathway.

To dissect mechanisms by which B4GALT1 regulates T-cell functions, we sorted OT-I T-cells after co-culture with target cells for genome-wide transcriptional analysis. Whole-genome RNA sequencing analysis confirmed the enhanced TCR activation in *B4galt1* knockout CD8[+] T-cells (*Figure 3h–j*). GSEA of DEGs (differentially expressed genes) between control and *B4galt1* gRNA-infected CD8[+] T-cells revealed that TCR signaling pathway was at the top of significantly altered pathways (*Figure 3j*).

When transplanted into wild-type mice with B16F10-OVA cells inoculated subcutaneously, *B4galt1* gRNA-infected OT-I T-cells showed significantly higher tumor killing activity than control gRNA-infected cells (*Figure 4a–c*). Analysis of tumor-infiltrated lymphocytes (TILs) demonstrated more infiltrated OT-I T-cells in tumors when *B4galt1* gRNA was infected (*Figure 4d* and *Figure 4—figure supplement 1*). Mechanistically, B16F10-OVA tumors had similar numbers of infiltrated *B4galt1* gRNA-infected OT-I T-cells as control gRNA-infected cells 24 hours after intravenous injection (*Figure 4—figure supplement 2a*), suggesting that B4GALT1 has no significant effect on initial tumor infiltration of OT-I T-cells. On the other hand, CFSE (carboxyfluorescein succinimidyl ester) analysis at a later time point (6 days after infusion) showed increased OT-I T-cell proliferation within tumors when B4GALT1 was inactivated (*Figure 4—figure supplement 2b*). Altogether, these results suggest that inhibition of N-galactosylation could enhance functions of CD8[+] T-cells both in vitro and in vivo, and that B4GALT1 could be a potential target to modulate activity of T-cells.

## Systematic identification of direct substrates of B4GALT1 on T-cell surface

To dissect the molecular mechanism by which B4GALT1 regulates T-cell activation, we analyzed N-glycome on OT-I T-cell surface by FACS staining (*Figure 5—figure supplement 1a*). Biotin-labeled *Erythrina cristagalli* lectin (ECL) and succinyl-wheat germ agglutinin (sWGA) were used to profile surface expression of terminal βGal and βGlcNAc, respectively. *B4galt1* knockout OT-I T-cells showed slightly but significantly decreased ECL staining and significantly increased sWGA staining, compared with control T-cells (*Figure 5—figure supplement 1b, c*). FACS staining with biotin-labeled recombinant Galectin-1 (Gal-1) showed a more dramatic difference between control and *B4galt1* knockout OT-I T-cells (*Figure 5—figure supplement 1d*).

To identify potential substrates of B4GALT1 on CD8[+] T-cell surface, we used a recombinant Gal-1 affinity column and lactose elution to purify N-glycosylated proteins from whole membrane-protein extracts of OT-I CD8[+] T-cells (*Figure 5a*). LC-MS analysis of proteins that were significantly different between control and *B4galt1* knockout OT-I T-cells revealed that both TCRα/β (OT-I) and CD8α/β were among the top list (*Figure 5b*). KEGG analysis also showed a significant enrichment of TCR signaling pathway (*Figure 5c*). We could verify reduced Gal-1 pull-down for CD8β and most of the other hits

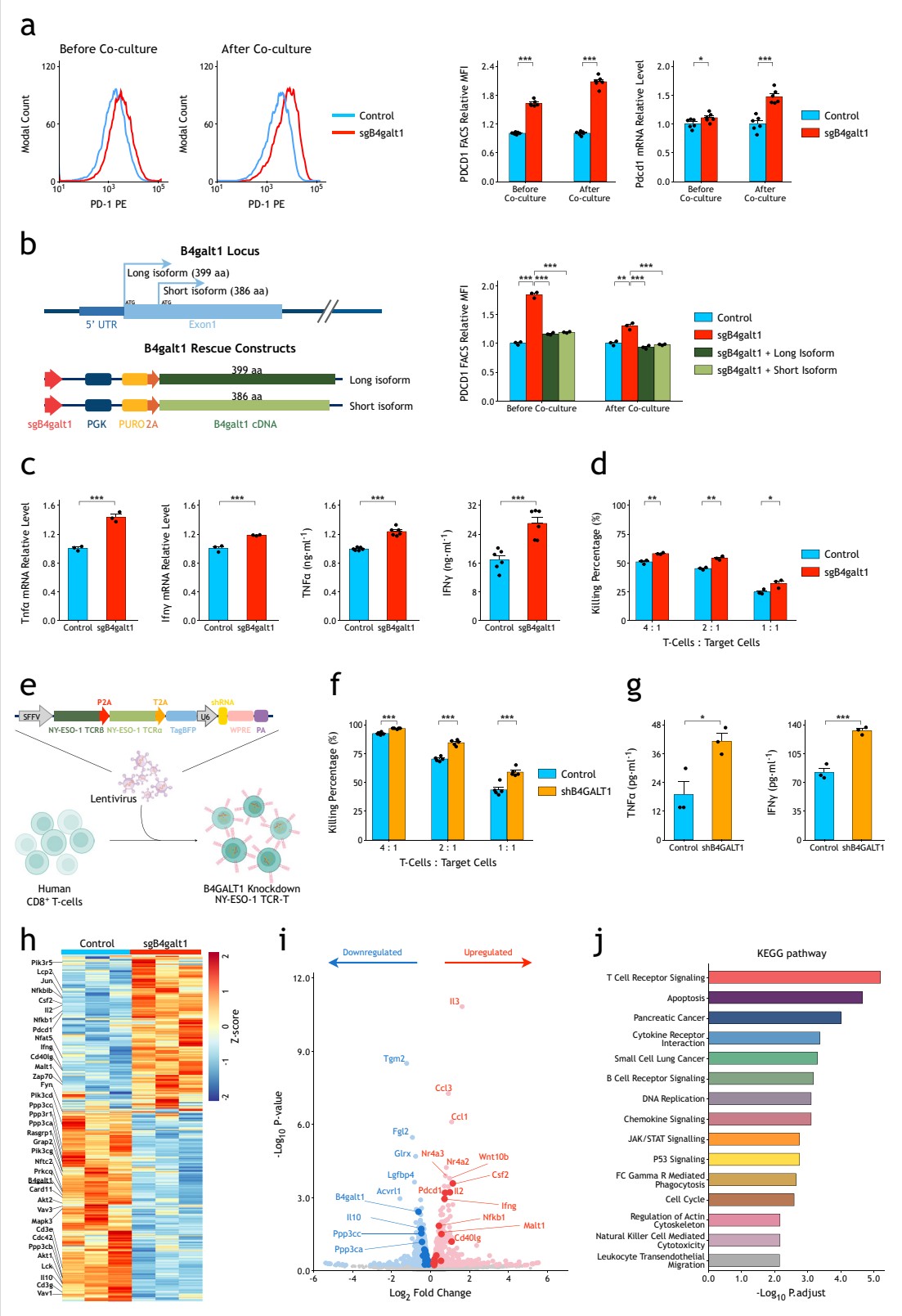

**Figure 3.** B4GALT1 suppression in CD8+ T-cells activates TCR signaling and enhances T-cell functions. (**a**) CRISPR/Cas9 knockout of *B4galt1* (sgB4galt1) (sg2) in CD8+ T-cells increases expression of PD-1 before and after co-culture with B16F10-OVA cells. The MFIs of PD-1 were measured by FACS (n=6). The relative mRNA levels of PD-1 were measured by quantitative RT-qPCR (n=6). The p-values were calculated using a two-tailed Student's *t*-test. (**b**) The effect of *B4galt1* knockout on PD-1 surface expression could be rescued by overexpression of either long- or short-isoform B4galt1 (n=3). The

*Figure 3 continued*

p-values were calculated using a two-tailed Student's *t*-test. (**c**) CRISPR/Cas9 knockout of *B4galt1* in CD8⁺ T-cells increases expression of TNFα and IFNγ after co-culture with B16F10-OVA cells. The relative mRNA levels were measured by quantitative RT-qPCR (n=3). The secreted TNFα and IFNγ in medium were measured by ELISA (n=6). The p-values were calculated using a two-tailed Student's *t*-test. (**d**) CRISPR/Cas9 knockout of *B4galt1* in OT-I CD8⁺ T-cells increases in vitro specific killing activities on B16F10-OVA cells (n=3). The p-values were calculated using a two-tailed Student's *t*-test. (**e**) Schematic view of B4GALT1 knockdown in human NY-ESO-1 TCR-T-cells. (**f**) Knockdown of B4GALT1 in human NY-ESO-1 TCR-T-cells by shRNA increases in vitro killing activities on A375 cells (n=5). The p-values were calculated using a two-tailed Student's *t*-test. (**g**) Knockdown of B4GALT1 in human NY-ESO-1 TCR-T-cells increases expression of TNFα and IFNγ after co-culture with A375 cells. The secreted TNFα and IFNγ in medium were measured by ELISA (n=3). The p-values were calculated using a two-tailed Student's *t*-test. (**h**) Heatmap demonstrating differentially expressed genes (DEGs) between B4galt1 knockout and control mouse OT-I CD8⁺ T-cells after co-culture. The genes in TCR signaling pathway are labeled on the left side. (**i**) Volcano plot showing upregulated and downregulated genes (p-value <0.01) in B4galt1 knockout mouse OT-I CD8⁺ T-cells after co-culture. The genes in TCR signaling pathway are labeled with dark blue and dark red. Top genes and some genes in TCR signaling pathway are annotated. The p-value was calculated using the Wald test, and p.adjust was calculated using Benjamini–Hochberg with the R package DESeq2 (version 1.22.2). (**j**) Bar graph showing KEGG pathways significantly changed in *B4galt1* knockout mouse OT-I CD8⁺ T-cells after co-culture. The p-value was calculated using the clusterProfiler (version 3.12.0) R package. All of these functional effects were biologically replicated at least twice. Data are shown as the mean ± SEM. *p<0.05; **p<0.01; ***p<0.001.

The online version of this article includes the following figure supplement(s) for figure 3:

**Figure supplement 1.** Effect of B4GALT1 knockout on functions of hCD19-CAR-T-cells and anti-CD3/28 stimulated T-cells.

**Figure supplement 2.** Effect of B4GALT1 knockout on functions of anti-CD3/28 stimulated T-cells.

---

in *B4galt1* knockout T-cells by western blotting with commercially available antibodies (*Figure 5d*). Interestingly, migration of CD8β in SDS-PAGE was significantly different between control and *B4galt1* knockout T-cell extracts, suggesting that CD8β is a direct substrate of B4GALT1 (*Figure 5d*). Indeed, treating whole membrane-protein extracts with PNGase F to remove all N-linked glycosylation on proteins omitted the migration difference of CD8β and most of other hits we identified (*Figure 5e*). These results suggest that B4GALT1 directly regulates N-glycosylation of cell surface proteins, such as components of TCR and CD8 complexes, on CD8⁺ T-cell.

It has been suggested that interaction between TCR and CD8 plays an important role for TCR activation (*Smith et al., 2018*; *Borger et al., 2014*). We hypothesized that aberrant galactosylation of TCR and CD8 might directly affect their interaction. For that, we used a fluorescence resonance energy transfer (FRET) assay to measure interaction between TCR and CD8 (*Smith et al., 2018*; *Borger et al., 2014*; *Figure 5—figure supplement 2*). As shown in *Figure 5f*, FRET signals of TCR-CD8 increased significantly in B4GALT1 deficient T-cells, compared with control T-cells. To confirm that reduced TCR-CD8 interaction is the major cause of TCR activation phenotypes in B4GALT1 knockout CD8⁺ T-cells, we generated a construct which fused the CD8β ectodomain (ECD) with CD3ε (*Figure 5g*). We expected that such fusion could artificially tether TCR with CD8 and bypass the regulation by B4GALT1. Indeed, overexpression of the CD8β-CD3ε fusion led to enhanced in vitro killing activities in control CD8⁺ T-cells. On the other hand, in B4GALT1-deficient CD8⁺ T-cells, such enhanced T-cell killing activities by fusion construct were significantly diminished (*Figure 5h*). Taken together, these results support a model that B4GALT1 directly regulates galactosylation of TCR complex and co-receptors to regulate TCR-CD8 interaction, which is essential for TCR activation and functions of T-cells.

## The expression levels of B4GALT1 and tumor-infiltrated CD8⁺ T-cells in tumor microenvironment are associated with prognosis of human patients

To investigate potential clinical relevance of B4GALT1 in human tumor patients, we analyzed cancer samples in The Cancer Genome Atlas (TCGA) (*Weinstein et al., 2013*). While expression levels of B4GALT1 are not linked to an overall survival benefit using data of all TCGA collected primary cancer samples, after normalized to expression level of CD8A, Kaplan–Meier curve showed a significantly better survival duration in patients with low expression of B4GALT1 (*Figure 6—figure supplement 1a, b*; *Tang et al., 2019*). In adrenocortical carcinoma (ACC), acute myeloid leukemia (LAML), lung adenocarcinoma (LUAD), and rectum adenocarcinoma (READ) data sets, for patients with higher expression of CD8A, low B4GALT1 expression level is significantly associated to better overall survival, compared with patients with lower expression of CD8A (*Figure 6a* and *Figure 6—figure supplement 1c*). On the contrary, in patients with lower expression of B4GALT1, high CD8A expression level is

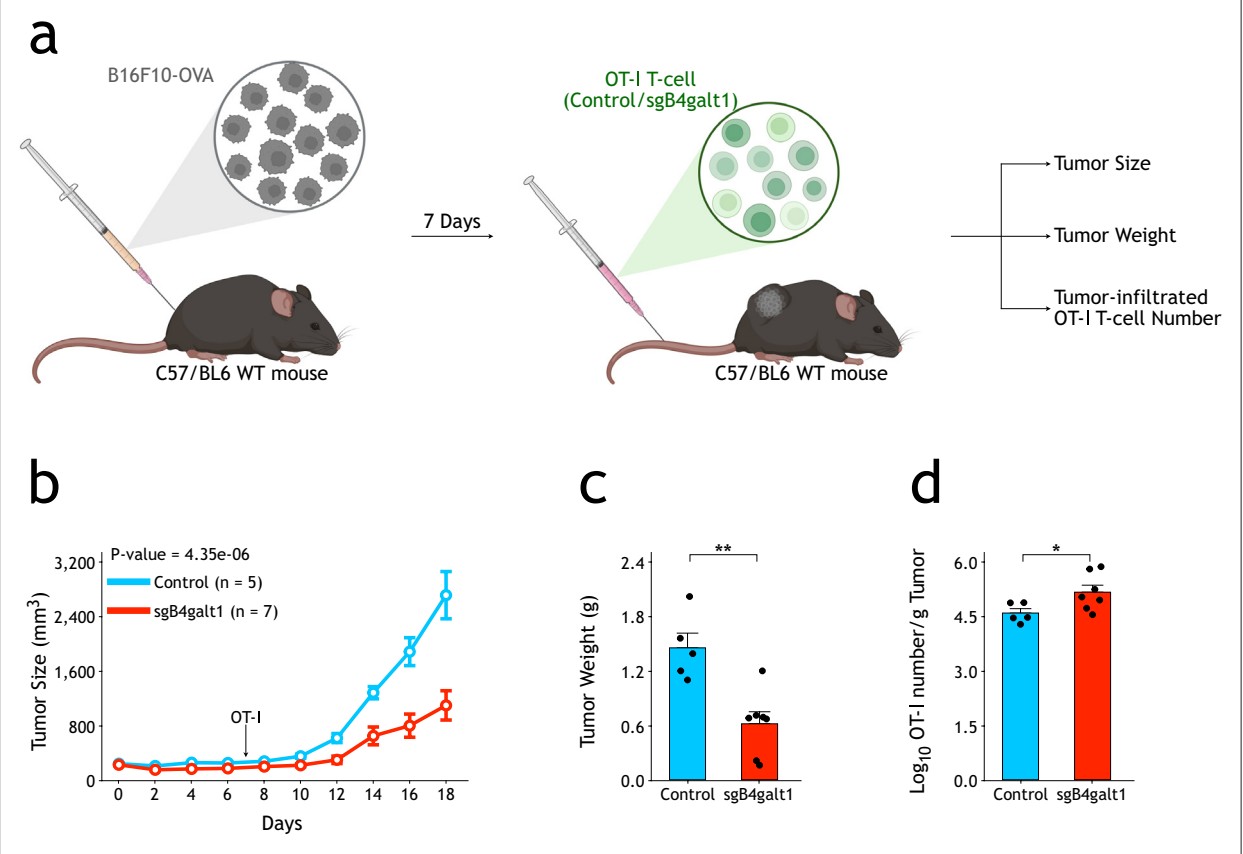

**Figure 4.** Knockout of *B4galt1* in CD8[+] T-cells enhances T-cell-mediated tumor immunotherapy. (**a**) Schematic view of *B4galt1* functional test in tumor microenvironment. (**b**) CRISPR/Cas9 knockout of B4galt1 in OT-I T-cells enhances growth control of B16F10-OVA tumors in vivo. The p-value was calculated using two-way ANOVA. (**c**) Compared with control OT-I T-cells, the tumors were significantly smaller when B4galt1 knockout OT-I T-cells were transplanted (n=5 for control, n=7 for sgB4galt1). The p-value was calculated using a two-tailed Student's *t*-test. (**d**) CRISPR/Cas9 knockout of B4galt1 increases numbers of OT-I T-cells in B16F10-OVA tumors (n=5 for control, n=7 for sgB4galt1). The p-value was calculated using a two-tailed Student's *t*-test. The in vivo functional effects were biologically replicated at least twice. Data are shown as the mean ± SEM. *p<0.05; **p<0.01.

The online version of this article includes the following figure supplement(s) for figure 4:

**Figure supplement 1.** Flow cytometry gating strategy for analysis of tumor-infiltrated OT-I T-cells.

**Figure supplement 2.** Effect of B4GALT1 knockout on functions of tumor-infiltrated OT-I T-cells.

significantly associated to better overall survival, compared with patients with higher expression of B4GALT1 (*Figure 6b* and *Figure 6—figure supplement 1c*). Collectively, these data suggest expression of B4GALT1 in tumor microenvironment and presence of tumor-infiltrated CD8[+] T-cells are jointly associated with prognosis of cancer patients.

## Discussion

Cytotoxic CD8[+] T-cells, which directly kill tumor cells, are key effector cells within tumor microenvironment (*Pan and Cheng, 2023*; *Coulie et al., 2014*). Following TCR activation by MHC-peptide complexes, PD-1 expression in T-cells is significantly upregulated (*Keir et al., 2008*; *Baumgaertner et al., 2022*). Cytokines, such as IL-2, IL-7, and interferons secreted by T-cells, can also induce high PD-1 expression (*Kinter et al., 2008*). Beyond its role as one of the markers for TCR activation, PD-1 suppresses further T-cell activation after binding to its ligand PD-L1, thereby preventing excessive immune responses (*Dong et al., 1999*). Therapeutically targeting PD-1 molecules on immune cells, especially cytotoxic CD8[+] T-cells, is one of the most successful strategies to treat tumors (*Topalian et al., 2012*). Understanding the mechanisms that regulate PD-1 expression on cytotoxic CD8[+] T-cells

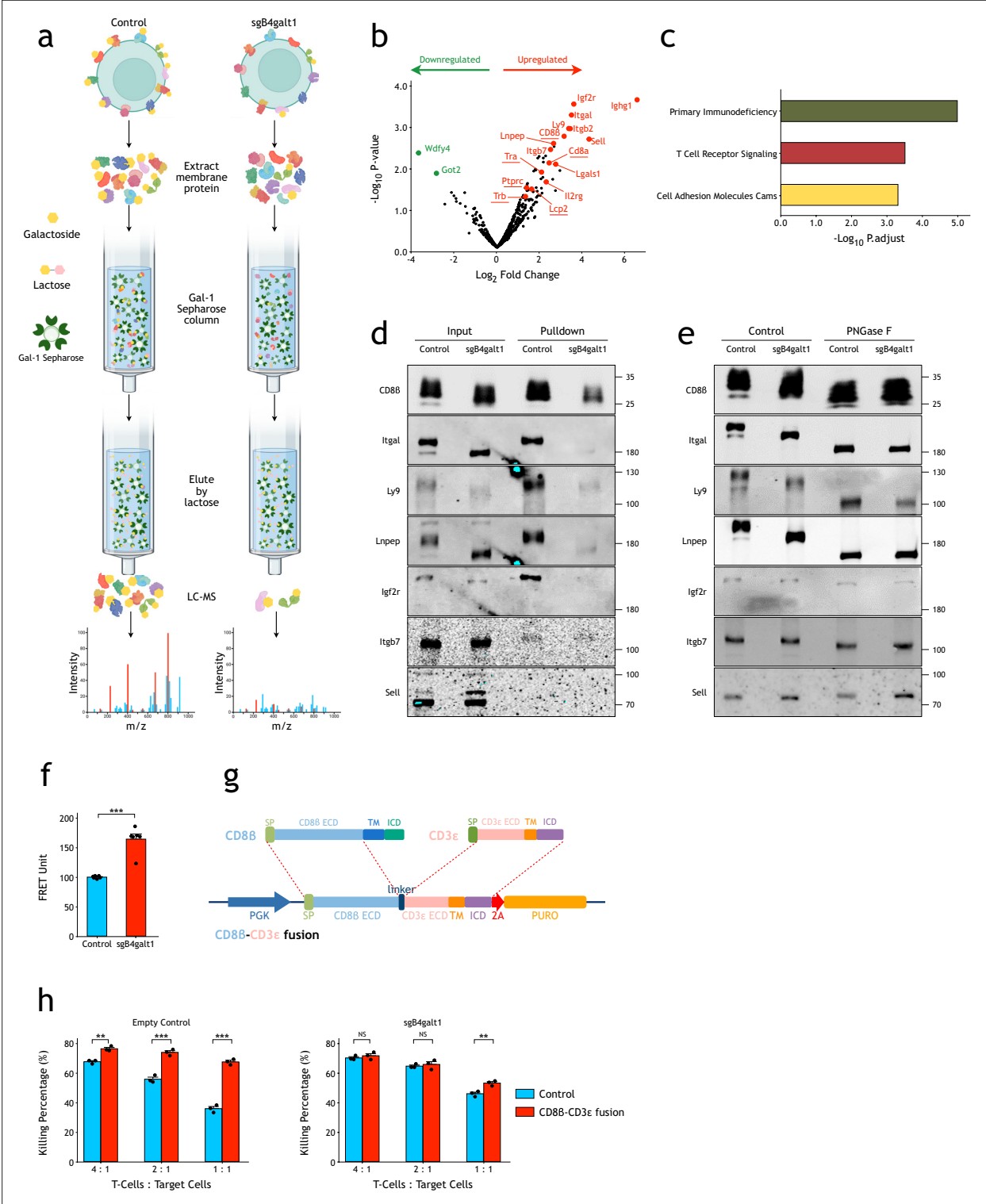

**Figure 5.** Systematic identification of direct substrates of B4GALT1 on T-cell surface. (**a**) Schematic view of recombinant Gal-1 pulldown and LC-MS experiments. (**b**) Volcano plot showing identified Gal-1 binding proteins in control and B4galt1 knockout OT-I cells. Proteins among the top list were annotated and labeled with red (decreased in *B4galt1* knockout) and green (increased in *B4galt1* knockout). Proteins in the TCR signaling pathway are underlined. The p-values were calculated using Limma in DEqMS (V1.8.0). (**c**) Bar graph showing KEGG pathways significantly changed in *B4galt1* knockout OT-I T-cells. The p-value was calculated using the clusterProfiler (version 3.12.0) R package. (**d**) Western blot verification of pulldown hits in top list. (**e**) N-glycome analysis with PNGase F suggests that CD8β is a direct substrate of B4GALT1. (**f**) Compared with wild-type control, B4GALT1 knockout OT-1 T-cells showed significantly stronger TCR-CD8 FRET signals (n=6). The FRET assays were biologically replicated three times. (**g**) Schematic view of

*Figure 5 continued on next page*

*Figure 5 continued*

the CD8β-CD3ε fusion construct. (**h**) Overexpression of a CD8β-CD3ε fusion protein bypassed the effect of B4GALT1 on T-cell in vitro killing activities (n=3). The killing assays were biologically replicated three times. All of the p-values were calculated by a two-tailed Student's *t*-test. Data are shown as the mean ± SEM. *p<0.05; **p<0.01; ***p<0.001; NS, not significant.

The online version of this article includes the following source data and figure supplement(s) for figure 5:

**Source data 1.** Original files for western blot result displayed in *Figure 5*.

**Source data 2.** PDF file containing original western blots for *Figure 5*, indicating the relevant bands and treatments.

**Figure supplement 1.** CRISPR/Cas9 knockout of B4GALT1 in OT-I T-cells alters surface-binding of lectins and galectin-1.

**Figure supplement 2.** Flow cytometry gating strategy for TCR-CD8 FRET assay.

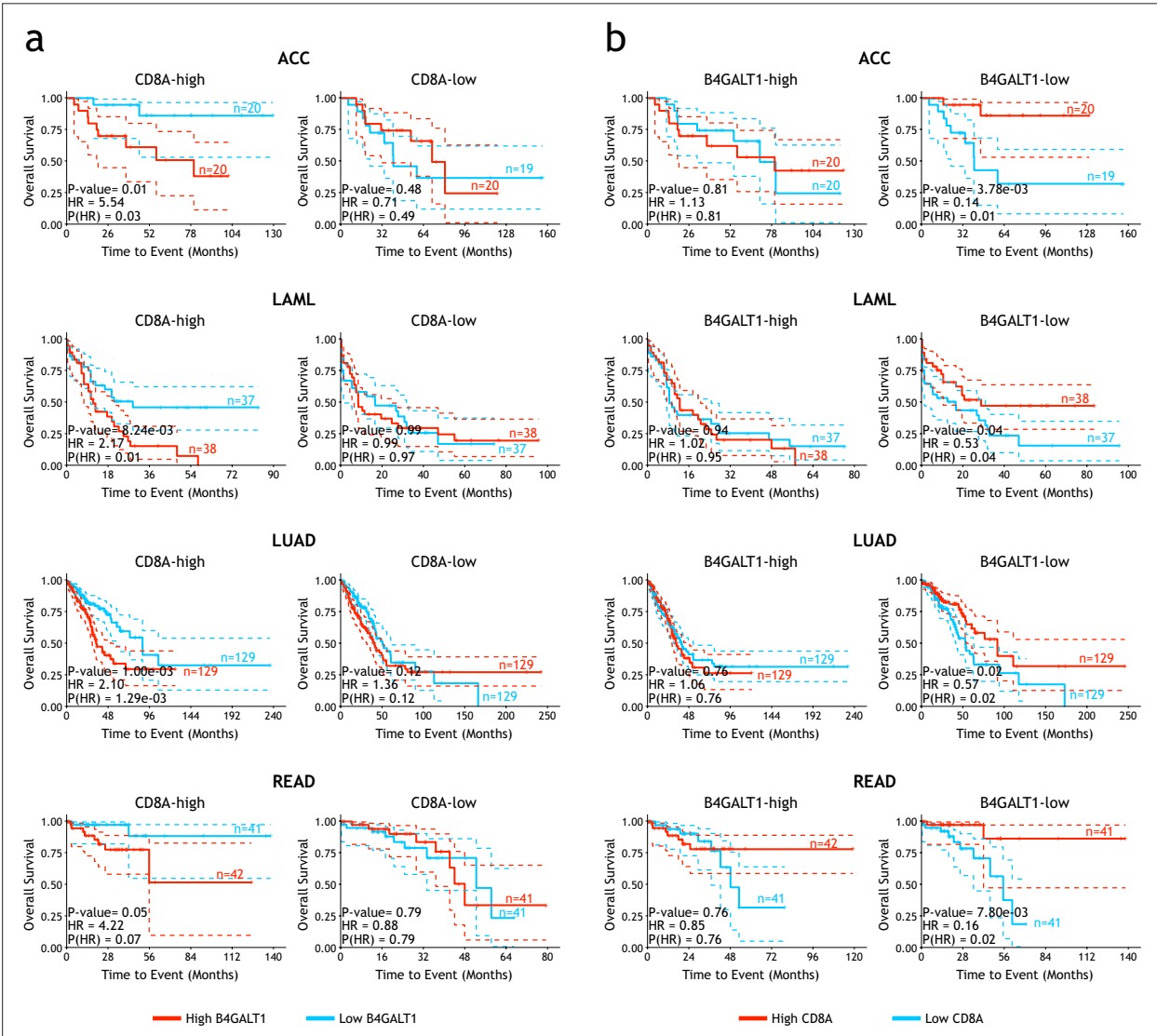

**Figure 6.** The expression levels of B4GALT1 and tumor-infiltrated CD8+ T-cells in the tumor microenvironment are associated with prognosis of human patients. (**a**) The association between B4GALT1 expression levels and overall survival for patients with different CD8A levels in TCGA-ACC, -LAML, -LUAD, and -READ cohorts. (**b**) The association between CD8A expression levels and overall survival for patients with different B4GALT1 levels in TCGA-ACC, -LAML, -LUAD, and -READ cohorts. The p-values for all survival curves were calculated using two-sided Log-rank test.

The online version of this article includes the following figure supplement(s) for figure 6:

**Figure supplement 1.** Expression levels of B4GALT1 and tumor-infiltrated CD8+ T-cells in tumor microenvironment are associated with prognosis of human patients.

presents important clues to optimize current tumor immunotherapy strategies, including CAR-T/TCR-T and immune checkpoint inhibitors, especially anti-PD-1 and anti-PD-L1 blocking antibodies.

By genome-wide CRISPR/Cas9 screenings, we systematically identified genes and pathways regulating PD-1 expression in primary CD8[+] T-cells. In such an ex vivo T-cell activation system, PD-1 serves mainly as an early activation marker for TCR. It will be interesting to adapt T-cell exhaustion culture with current screenings, where PD-1 is induced as an exhaustion marker (*Jiang et al., 2015*; *Kamphorst et al., 2017*; *Sharpe and Pauken, 2018*). The results that *Pdcd1* gene and some previously reported regulators of PD-1, such as *Satb1* and *Fut8*, were successfully identified demonstrate the effectiveness of this screening system. Interestingly, genes in N-glycan biosynthesis are significantly enriched to negatively regulate PD-1 expression in CD8[+] T-cells. Inhibition of B4GALT1 could activate expression of PD-1 and functions of CD8[+] T-cells both in vitro and in vivo. Mechanistically, B4GALT1 modulates galactosylation of proteins on CD8[+] T-cell surface, including proteins in TCR complex and its co-receptors. The decreased galactosylation of TCR complex and CD8 enhances TCR-CD8 interaction, which is the major downstream mechanism that B4GALT1 regulates TCR activation. Our results not only demonstrate the important roles of protein N-glycosylation in regulating functions of CD8[+] T-cells but also prove that B4GALT1 is an effective target for tumor immunotherapy. In particular, previous studies have demonstrated the roles of B4GALT1 in cancer cells such as in LUAD and CRC (*Cui et al., 2023*; *Hsu et al., 2024*; *De Vitis et al., 2019*). Our study suggests that inhibition of B4GALT1 in both cancer cells and CD8[+] T-cells may synergistically suppress tumor growth.

Based on results from individual studies of specific glycosidases and glycosyltransferases, N-linked glycosylation of T-cell surface proteins has been proposed to regulate T-cell development, activation, and functions (*Abdelbary and Nolz, 2023*). Here, we identified roles of N-glycome synthesis in regulating PD1 expression of CD8[+] T-cells by unbiased genome-wide screenings. To further investigate mechanisms, we systematically characterized substrates of B4GALT1 on CD8[+] T-cell surface, suggesting the model that B4GALT1 regulates TCR activation by directly modulating N-glycosylation of components of TCR and its co-receptor complexes. In addition, several interesting observations also provide mechanistic insights supporting our hypothesis. Knockout of B4GALT1 only significantly affects activation of exogenous and exogenous TCR (OT-I and NY-ESO-1, respectively) but not CAR (hCD19-CAR) in CD8[+] T-cells. Similarly, knockout of CD8a shows different effects on TCR and CAR activation (data not shown). Moreover, different from results obtained after co-culture with target cells, when B4GALT1 knockout CD8[+] T-cells were stimulated by anti-CD3/28 only, they did not show significantly enhanced activation. Further studies are necessary to clarify whether B4GALT1-mediated N-galactosylation modifications directly or indirectly affect functions of its other substrates and how cancer cells contribute to activation of CD8[+] T-cells in a B4GALT1-dependent manner.

Adoptive T-cell therapy has been actively implicated for treatment of cancer and chronic infections (*Kalos and June, 2013*; *Kamphorst and Ahmed, 2013*). While CAR-T and TCR-T therapies demonstrated promising effects in controlling hematological malignances, their applications in solid tumors remain challenging (*Sterner and Sterner, 2021*; *Zhang et al., 2024*). How to enhance activities of in vitro expanded T-cells is also a major barrier for tumor-infiltrating lymphocyte (TIL) therapy (*Wu et al., 2012*). Our in vitro and in vivo results strongly support that B4GALT1 is a potential target to enhance the efficacy of adoptive T-cell therapy.

In summary, our study utilized unbiased genome-wide and custom small library screenings to identify PD-1 regulators in CD8[+] T-cells and discovered a fundamental role for a gene involved in N-glycan biosynthesis—B4GALT1, which can enhance T-cell activation and functions both in vitro and in vivo. Our findings highlight the power of protein N-glycosylation in regulating functions of CD8[+] T-cells and suggest that B4GALT1 is a potential target for tumor immunotherapy. It will provide a new perspective and direction for the research of T-cell regulators to enhance the efficacy of tumor immunotherapy.

## Declarations

### Ethics approval and consent to participate

All animal experiments were conducted following the Ministry of Health national guidelines for housing and care of laboratory animals and performed in accordance with institutional regulations after review and approval by the Institutional Animal Care and Use Committee at the National Institute of Biological Sciences and Chinese Institutes for Medical Research. The assigned approval/accreditation number: AEEI-2023-223.

This manuscript does not report on or involve the use of any human data or tissue.

## Consent for publication

This manuscript does not contain data from any individual person.

# Materials and methods

## Cell lines

HEK293T cell line was purchased from ATCC (CRL-3216). B16F10 cell line (ATCC: CRL-6475) was provided by the laboratory of Dr. Ting Chen (National Institute of Biological Sciences, Beijing). Nalm6 cell line (ATCC: CRL-3273) was provided by the laboratory of Dr. Zhaoqing Ba (National Institute of Biological Sciences, Beijing). A375 cell line (ATCC: CRL-1619) was provided by the laboratory of Dr. Feng Shao (National Institute of Biological Sciences, Beijing). All of these cell lines' authentication services were offered by ATCC. B16F10-OVA cell line was constructed by infecting B16F10 cells with lentivirus encoding chicken ovalbumin. All cell lines were routinely tested and negative for myco-plasma contamination.

B4galt1 rescue vectors were constructed based on pMSCV-U6 sgB4galt1-PGK-puro-2A-BFP vector. Long isoform (UniProt, P15535-1) and short isoform (UniProt, P15535-2) were amplified from cDNA of mouse T-cells and used to replace the BFP fragment by seamless cloning kit (Biomed, CL116).

The CD8β-CD3ε fusion vector was constructed by fusing CD8β ectodomain (ECD) and its signal peptide with a linker region and the full-length of CD3ε, based on pMSCV-U6 sgB4galt1/empty-PGK-puro-2A-BFP vector mentioned above.

HEK293T, B16F10, and A375 cells were cultured in DMEM (Gibco C11965500BT) supplemented with 10% fetal bovine serum (FBS), 2 mmol L-glutamine, 100 μg/ml penicillin, and 100 U/ml strep-tomycin (all purchased from Gibco). Nalm6 cells were maintained in RPMI-1640 (HyClone) medium supplemented with 15% FBS, 100 μg/ml penicillin, 100 U/ml streptomycin, 10 mM HEPES, 0.1 mM 2-mercaptoethanol, 2 mM L-glutamine, and 1xMEM nonessential amino acids.

## Retrovirus and lentivirus preparation

Retrovirus was packaged by co-transfecting HEK293T cells with MSCV vector and pCL-Eco. 48 hours post-transfection, supernatant was collected and filtered through a 0.45 μm filter to remove cell debris. The virus was then concentrated by centrifugation in Beckman Optima L-100XP (SW32Ti) at 25,000 rpm (107,000 × $g$) for 2.5 hours at 4°C and resuspended in RPMI-1640 basic medium.

Lentivirus was packaged by co-transfecting HEK293T cells with lentiviral vector, psPAX2, and pMD2.G. 48 hours post-transfection, supernatant was collected and filtered through a 0.45 μm filter to remove cell debris. Virus was used for infection directly or concentrated by centrifugation at 25,000 rpm (107,000 × $g$) for 2.5 hours at 4°C and resuspended in PBS.

To produce lentivirus for human CD8[+] T-cell infection, HEK293T cells were seeded in Opti-MEM I Reduced Serum Medium (Gibco, 31985-070) supplemented with 5% FBS, 1 mM sodium pyruvate (Gibco, 11360-070), and 1xMEM nonessential amino acids (Gibco, 11140-050) the day before trans-fection. 6 hours post-transfection, the transfection medium was replaced with fresh medium supple-mented with 1x ViralBoost (Alstem Bio, VB100). Lentivirus was collected and filtered through a 0.45 μm filter 24 hours and 48 hours after transfection separately, followed by addition of Lenti-X-Concentrator (Takara, 631232). Lentivirus was concentrated following manufacturer's instructions and resuspended in X-VIVO 15 medium (Lonza, 04-418Q) in 1% of the original volume.

## Animals

Female C57BL/6J mice were purchased from Beijing Vital River Laboratory Animal Technology Co., Ltd. OT-I mice were provided by the laboratory of Dr. Hai Qi (Tsinghua University). Cas9-EGFP/OT-I mice were generated by breeding OT-I and Cas9-EGFP knock-in mice (*Platt et al., 2014*) at the animal facility of the National Institute of Biological Sciences, Beijing. Six- to eight-week-old mice were used at the start of experiments. Animals were housed under specific pathogen-free conditions in individu-ally ventilated cages in a controlled 12-hour day-night cycle with standard food and water provided ad libitum. All animal experiments were conducted following the Ministry of Health national guidelines for housing and care of laboratory animals and performed in accordance with institutional regulations

after review and approval by the Institutional Animal Care and Use Committee at the National Institute of Biological Sciences.

## Retroviral whole-genome CRISPR/Cas9 gRNA library construction

A whole-genome CRISPR knockout gRNA library (1000000096) was purchased from Addgene. The gRNA regions were PCR amplified with primer pair F:5'-ggctttatatatcttgtggaaaggacgaaacaccg-3' and R:5'-ctagccttattttaacttgctatttctagctctaaaac-3', and then transferred into MSCV-gRNA-PGK-PURO-2A-BFP vector by Gibson reaction. For the knockout of individual genes, a single gRNA was cloned into the same vector by Gibson reaction.

## Retroviral small custom gRNA library construction

A total of 1398 genes were selected according to whole-genome knockout screening results. On average, three gRNAs were selected from the initial library for each gene according to gRNA performance (*Supplementary file 3*). A total of 105 intergenic control gRNAs were also added. Oligonucleotides containing the guide sequence were synthesized (Custom Array), PCR-amplified, and cloned into MSCV-gRNA-PGK-PURO-2A-BFP vector via Gibson reaction.

## Ex vivo T memory cell culture, infection, and adoptive transfer

On day 0, splenocytes were prepared from 6- to 8-week-old Cas9-EGFP/OT-I female mice and cultured with IL2 (10 ng/ml) and SIINFEKL peptide (10 ng/ml) in complete RPMI media (RPMI 1640, 10% FBS, 20 mM HEPES, 1 mM sodium pyruvate, 0.05 mM 2-mercaptoethanol, 2 mM L-glutamine, 100 U/ml streptomycin, and 100 µg/ml penicillin) at a density of $1 \times 10^6$ cells/ml for 24 hours. On day 1, activated Cas9-EGFP/OT-I T-cells were enriched by Percoll isolation as previously described (*Kurachi et al., 2017*) and spin-infected ($2000 \times g$, 30°C, 1 hour, with minimum acceleration and no brake) with retrovirus supplemented with polybrene (8 µg/ml) in 24-well plate. After spin-infection, the plate was placed into a $CO_2$ incubator at 37°C for 5 hours and cultured with IL2 (2 ng/ml), IL7 (2.5 ng/ml), and IL15 (10 ng/ml) in complete RPMI media at a density of $3 \times 10^5$ cells/ml. Two days after infection (day 3), cells were selected by 3 µg/ml puromycin in the presence of IL2/IL7/IL15 for another 4 days. On day 7, cells were used for co-culture experiments or adoptive transfer.

## Human T-cell isolation, culture, and transduction

Human peripheral blood mononuclear cells (PBMCs) were acquired from healthy donors. CD8+ T-cells were isolated using EasySep negative selection kit (STEMCELL, Cat#17953), and stimulated with anti-CD3/CD28 Dynabeads (Thermo Fisher Scientific, Cat#40203D). T-cells were cultured in X-VIVO medium with 10% FBS, 2 mmol L-glutamine, 100 µg/ml penicillin, 100 U/ml streptomycin, and human IL2 (500 IU/ml). 48 hours after activation, T-cells were transduced in a lentivirus-coated plate, centrifuged at $1200 \times g$ for 90 minutes at 37°C. Lentivirus transduction was repeated once 24 hours later. Positive transduced cells were sorted 3 days later. During T-cell expansion, the cells were maintained at a concentration of $3 \times 10^5$ cells/ml.

## Anti-CD19-CAR and NY-ESO-1-specific TCR-T-cell generation

pMSCV-CD19 scFv (FMC63)-IRES-RFP-U6 sgRNA vector was constructed from a pMSCV-CD19 scFv-IRES-RFP vector provided by laboratory of Dr. Feng Shao (National Institute of Biological Sciences, Beijing). An U6 sgRNA cassette was assembled by seamless cloning kit (Biomed, CL116).

CD8+ T-cells were purified from splenocytes of Cas9-EGFP knock-in mice by a mouse CD8+ T-cell isolation kit (R&D, MAGM203). At day 0, CD8+ T-cells were stimulated with anti-CD3 (1 µg/ml, Cat#14-0031-86) and anti-CD28 (0.5 µg/ml, Cat#102116) in complete RPMI 1640 medium containing 20 ng/ml IL2 for 24 hours. On day 1, activated CD8+ T-cells were enriched by Percoll isolation and spin-infected ($2000 \times g$, 30°C, 1 hour, with no acceleration or brake) with retrovirus expressing anti-CD19-CAR and sgRNA supplemented with polybrene (8 µg/ml) in 24-well plate. The RFP+ cells were sorted on day 3 for further culture. CD8+ T-cells were cultured with the same condition as T memory cells. In vitro killing assay was performed on day 7 by mixing anti-CD19-CAR-T-cells with Nalm6.

NY-ESO-1 specific TCR (1G4) (*Robbins et al., 2008*) was synthesized and cloned into a lentiviral backbone with a SFFV promoter. T2A-BFP fragment and U6 shRNA cassette were assembled by seamless cloning kit (Biomed, CL116) simultaneously.

NY-ESO-1 specific TCR-T-cells were generated following the human CD8[+] T-cell isolation, culture, and transduction protocols. 3 days post-transduction, BFP[+] cells were sorted for further culture. 7 days post-transduction, in vitro killing assay was performed by mixing T-cells with A375 cells.

## CRISPR/Cas9 screening

For ex vivo PD-1 expression screening, at day 6, $3 \times 10^6$ B16F10-OVA cells were plated in 15 cm dishes in DMEM (DMEM +10% FBS +Pen/Strep). On day 7, puromycin-selected OT-I T-cells were resuspended in a final concentration of $6 \times 10^5$ cells/ml with complete RPMI 1640 medium supplemented with IL2 (2 ng/ml), IL7 (2.5 ng/ml), and IL15 (10 ng/ml), and added to B16F10-OVA cells at a T-cell:B16F10-OVA cell ratio of 4:1. Cells were co-cultured overnight at 37°C. CD8[+] T-cells were then collected and stained with PE anti-PD-1 in 2% FBS/PBS for 30 minutes on ice. The highest and lowest 5% PD-1[+] cells were sorted via BD FACSAria.

For the secondary small custom library in vivo screening, following 4-day puromycin selection, an aliquot of $2 \times 10^6$ infected OT-I T-cells was saved as input (approximately 269X cell coverage per sgRNA). Library-infected OT-I T-cells ($2 \times 10^6$ cells per recipient) were intravenously transferred into mice bearing day 14 B16F10-OVA tumors. In total, 24 mice were used as recipients. At 7 days post-adoptive transfer, B16F10-OVA tumors were digested into single-cell suspensions, and tumor-infiltrated CD8[+] T-cells were isolated by biotin anti-mouse CD8a (Biolegend, 100704) and streptavidin beads. Meanwhile, a 1/20 volume aliquot of single-cell suspension was used for OT-I staining to estimate numbers of infiltrated OT-I T-cells in each tumor. A total of $1 \times 10^4$ to $1 \times 10^5$ OT-I T-cells were collected per tumor.

## Sequencing library preparation

Genomic DNA was extracted by phenol/chloroform extraction. Primary PCR was performed using Titanium Taq DNA Polymerase (Clontech, 639242) to amplify the sgRNA region. A secondary PCR was performed to add adaptors and indexes to each sample. Nova-seq 150 bp paired-end sequencing (Illumina) was performed.

## Analysis of screening results

Raw reads were preprocessed using sequence-grooming tools to remove adaptor sequences with Cutadapt (version 3.4) (*Martin, 2011*) and to merge reads with FLASH (version 1.2.11) (*Magoč and Salzberg, 2011*). Then, MAGeCK (version 0.5.9.5) (*Li et al., 2014*) was used to analyze the screening data. Specifically, the MAGeCK 'count' command was employed to generate read counts for all samples, which were subsequently merged into a count matrix. The MAGeCK 'test' command was then used to identify the top negatively and positively selected gRNAs or genes, using default settings. The software is available at https://sourceforge.net/projects/mageck/. Additionally, Gene Set Enrichment Analysis (GSEA) in the Kyoto Encyclopedia of Genes and Genomes (KEGG) functional pathway was performed using the GSEA() function from the R package clusterProfiler (version 3.12.0) (*Yu et al., 2012*), with default parameters. The KEGG database was selected from the "C2" category of the R package msigdbr (version 7.5.1), available at https://igordot.github.io/msigdbr/.

## T7 endonuclease I (T7E1) assay

The T7E1 cleavage assay was performed using a previously reported protocol (*Duan et al., 2014*). In brief, genomic DNA of infected OT-I T-cells was extracted and then subjected to PCR amplification of sgRNA targeting regions using primers indicated in *Supplementary file 3*. PCR products were gel-purified using StarPrep DNA Gel Extraction kit (GenStar, Cat#D205-04), and annealed in 1xPCR buffer (TaKaRa, Cat#9151A). The annealed products were incubated with five units of T7E1 (NEB, Cat#M0302L) at 37°C for 20 minutes and then analyzed by PAGE.

## Flow cytometry

For surface staining, cells were stained in 2% FBS/PBS on ice for 30 minutes. Intracellular staining was performed with a fixation/permeabilization kit (BD Biosciences) according to manufacturer's instructions. The following antibodies were used: APC anti-mouse CD8a (Invitrogen, 17-0081-83), PE anti-mouse CD279 (PD-1) (BioLegend, 109104), APC anti-mouse CD279 (PD-1) (BioLegend, 109112), PE-streptavidin (BioLegend, 405203), biotinylated erythrina cristagalli lectin (ECL) (B-1145-5),

biotinylated succinylated wheat germ agglutinin (sWGA) (B-1025S-5), biotinylated recombinant Gal-1, APC anti-human CD8a (BioLegend, 300912). Flow cytometry was performed by BD FACSAria and data were analyzed with FlowJo.

## ELISA

OT-I T-cells ($3 \times 10^5$/ml) were co-cultured with B16F10-OVA cells at 37°C for 8 hours in the presence of IL2 (2 ng/ml), IL7 (2.5 ng/ml), and IL15 (10 ng/ml). Supernatants were collected after co-culture. TNFα and IFNγ in the culture supernatant were measured using ELISA kits (ABclonal, Cat#RK00027, Cat#RK00019). Samples were plated in duplicate.

Human CD8$^+$ T-cells ($3 \times 10^5$ /ml) were co-cultured with A375 cells at 37°C for 24 hours in the presence of human IL2 (500 IU/ml). Supernatants were collected for measurement of TNFα and IFNγ secretion using ELISA kits (ABclonal, Cat#RK00030, Cat#RK00015).

## Mouse tumor models and adoptive transfer experiments

$4 \times 10^5$ B16F10-OVA cells were injected subcutaneously into female C57BL/6J mice. Seven days post-injection, mice bearing tumors of similar size were randomly separated into two groups. A total of $2 \times 10^6$ *B4galt1* gRNA-transduced or control gRNA-transduced OT-I T-cells were injected intravenously. Tumors were then measured every two days with an electronic digital caliper. Tumor volume was calculated as width × width × length × 1/2.

## Tumor-infiltrated lymphocyte (TIL) isolation

B16F10-OVA tumors were cut into small pieces in 6-well plates containing 5 ml RPMI 1640, 2% FBS, and 50 U/ml collagenase type IV (Invitrogen, V900893). Samples were incubated at 37°C for 1 hour with rotation. Suspensions were passed through a 70 µm strainer and washed three times with PBS. Samples were then used for antibody staining and FACS analysis.

## T-cell infiltration and proliferation within tumors

Following 4-day puromycin selection, infected OT-I T-cells were labeled with CFSE (carboxyfluorescein diacetate succinimidyl ester, Invitrogen), and $2 \times 10^6$ labeled cells were intravenously transferred into day 14 B16F10-OVA tumor-bearing mice. CFSE dilution was quantified by flow cytometry at 24 hours and day 6 following transfer.

## In vitro targeted cell killing

For OT-I T-cell killing activity assay, B16F10-OVA cells and B16F10 cells were labeled as CFSE$^{hi}$ (2.5 µM) and CFSE$^{lo}$ (50 nM), respectively, and then co-cultured with OT-I T-cells in 96-well plates at the indicated ratios. Cancer cells without the addition of OT-I T-cells were used as controls. Following 24 hours of incubation, the ratios of CFSE$^{hi}$ and CFSE$^{lo}$ populations were detected by FACS.

Specific killing was calculated by the following equation: specific killing percentage = [1-(CFSE$^{hi}$/CFSE$^{lo}$ of T-cell)/(CFSE$^{hi}$/CFSE$^{lo}$ of cancer cell only)]×100%.

For NY-ESO-1 TCR-T and hCD19-CAR-T-cell in vitro killing assays, A375 cells and Nalm6 cells were co-cultured with T-cells for 24 hours and 8 hours, respectively. The killing percentage was calculated by the following equation: killing percentage = [1-(survived target cell number / survived target cell number in non-T-cell wells)]×100%.

## Proteomic analysis of Gal-1 binding proteins in CD8$^+$ T-cells

Recombinant Gal-1 was purified (*Prato et al., 2020*) and coupled with Sepharose beads (Cat#GE17-0906-01). *B4galt1* knockout OT-I T-cells were co-cultured with MC38 for 8 hours and then stained by anti-Gal-1 antibody for sorting of Gal-1 negative OT-I T-cell population. Membrane extracts of OT-I T-cells were prepared by Mem-PER Plus Kit (Cat#89842Y). Extracts were incubated with Gal-1-Sepharose beads. After washing, binding proteins were eluted by 20 mM lactose and precipitated by DOC/TCA, and then separated on SDS-PAGE followed by silver staining (Sigma, PROTSIL1). The samples were digested and subjected to LC-MS (LTQ ORBITRAP Velos mass spectrometer, Thermo Fisher Scientific, San Jose, CA, USA).

Raw data was processed by Proteome Discoverer (version 1.4) to obtain a PSM (peptide spectrum matches) raw intensity table of all identified proteins for further analysis with R package DEqMS (*Zhu*

*et al., 2020*). Raw intensity values were log2 transformed, and for each PSM, the median of log2 intensity was subtracted to get a relative log2 ratio. Then a different expression of protein calculation was enabled in the eBayes() function with method receiver operating characteristic (ROC) analysis. Differentially expressed proteins were selected by criteria p<0.05. Those proteins were used for functional enrichment analysis with R package ClusterProfiler (version 3.12.0) (*Wu et al., 2021*).

## Western blot

10 µg cell membrane extracts of OT-I T-cells were separated by SDS-PAGE gels. Gels were transferred to nitrocellulose membranes (Amersham Protran, Cat#10600001). Anti-CD8b (Abcam, Cat# ab228965) (1:1000), Anti-Ly9 (Abcam, Cat# ab252931) (1:1000), anti-Igf2g (Sino Biological, Cat# 107533-T40) (1:500), anti-Itgal (Solarbio, Cat# K002895P) (1:500), anti-Itgb7 (Solarbio, Cat#K004272P) (1:500), anti-Lnpep (Santa Cruz, Cat#sc-365300) (1:100), and anti-Sell (Santa Cruz, Cat#sc-390756) (1:100) were used as primary antibodies. Secondary antibodies were IRDye 680RD Donkey anti-Mouse IgG (LI-COR, P/N 926-68072) (1:10,000), IRDye 800CW goat anti-mouse IgG (LI-COR, P/N 926-32210) (1:10,000), IRDye 680RD goat anti-rabbit IgG (LI-COR, P/N 926-68071) (1:10,000), IRDye 800CW goat anti-rabbit IgG (LI-COR, P/N 926-32211) (1:10,000). The membranes were scanned on an Odyssey imager (LI-COR).

## FRET

TCR-CD8 FRET was measured by flow cytometry (*Smith et al., 2018*). T-cells were incubated with PE-anti-Vα2 and APC-anti-CD8α at 4°C for 30 minutes. Samples were stained with either antibody, both, or neither. After staining, samples were fixed with fixation buffer at 4°C for 10 minutes. FRET emission was assessed and FRET efficiency was calculated in FRET units as reported previously (*Smith et al., 2018*).

## RT-qPCR and RNA-seq

Total RNA was extracted using TRIzol Reagent (Ambion, Cat# 15596018). cDNA was synthesized by the ImProm-II Reverse Transcriptase system (Promega, Cat# A3801) using 100 ng RNA per reaction. The qPCR reactions were prepared with TB Green Premix Ex Taq (Takara, Cat# RR420A) using 1 µl cDNA per reaction in a 20 µl reaction volume. The relative gene expression levels were normalized to GAPDH.

Total RNA was used as input material for the RNA sample preparations. Sequencing libraries were generated using NEBNext Ultra RNA Library Prep Kit for Illumina (NEB, Cat#E7530L) following manufacturer's recommendations and index codes were added to attribute sequences to each sample. Prepared libraries were quantified and sequenced by the Illumina NovaSeq 6000 S4 platform with 150 bp paired-end reads.

## RNA-seq data analysis

Raw data was aligned to mouse reference genome (mm10) with Ensemble version 98 gene annotation using TopHat2 (version 2.1.1). The raw count of each gene was then calculated using HTSeq (version 1.99.2) (*Putri et al., 2022*) and normalized to Fragments Per Kilobase of transcript per Million mapped reads (FPKM) by cufflinks (version 2.2.1) (*Trapnell et al., 2012*). Differential gene expression analysis was performed using the R-based toolkit DESeq2 (version 1.38.0) (*Love et al., 2014*), with differentially expressed genes (DEGs) defined by a significance threshold of p<0.05 or p<0.01. Functional enrichment analysis of these DEGs was conducted using the R package ClusterProfiler (version 3.12.0) (*Wu et al., 2021*).

## Statistical analysis

Statistical analyses were conducted using R 4.1.0. Unless otherwise stated, statistical significance was determined using a two-tailed paired Student's *t*-test (*p<0.05, **p<0.01, ***p<0.001; NS, not significant) with the function t.test(). Two-way ANOVA was performed using the function aov(). In the figures, the mean and standard error of the mean (SEM) are presented, with error bars representing the SEM value, calculated using the function sem().

## Acknowledgements

We thank members of YZ lab for helpful discussion and support. We thank the core facility center at Capital Medical University (CMU) for helpful assistance and service. We thank the municipal government of Beijing and the Ministry of Science and Technology (MOST) of China for funds allocated to NIBS, and the municipal government of Beijing for funds allocated to CIMR. The research in YZ lab is supported by grants from National Key R&D Program of China (2021YFA1101002), National Natural Science Foundation of China (81773304, 81572795), the 'Hundred, Thousand and Ten Thousand Talent Project' by Beijing municipal government (2019A39).

## Additional information

### Competing interests

Yu Hong, Xiaofang Si, Wenjing Liu, Xueying Mai, Yu Zhang: Part of this research has been submitted for a patent. Application number: 202380091410.9.

### Funding

| Funder | Grant reference number | Author |
|---|---|---|
| National Key Research and Development Program of China | 2021YFA1101002 | Yu Zhang |
| National Natural Science Foundation of China | 81773304 | Yu Zhang |
| National Natural Science Foundation of China | 81572795 | Yu Zhang |
| the "Hundred, Thousand and Ten Thousand Talent Project" by Beijing municipal government | 2019A39 | Yu Zhang |

The funders had no role in study design, data collection and interpretation, or the decision to submit the work for publication.

### Author contributions

Yu Hong, Xiaofang Si, Wenjing Liu, Data curation, Formal analysis, Supervision, Validation, Investigation, Visualization, Methodology, Writing – original draft, Project administration, Writing – review and editing; Xueying Mai, Data curation, Formal analysis, Supervision, Validation, Investigation, Visualization, Methodology, Writing – original draft, Writing – review and editing; Yu Zhang, Conceptualization, Resources, Data curation, Formal analysis, Supervision, Funding acquisition, Validation, Investigation, Visualization, Methodology, Writing – original draft, Project administration, Writing – review and editing

### Author ORCIDs

Yu Hong (iD) https://orcid.org/0009-0009-9899-7139
Xiaofang Si (iD) https://orcid.org/0009-0000-5076-1201
Wenjing Liu (iD) https://orcid.org/0009-0003-9286-3857
Xueying Mai (iD) https://orcid.org/0009-0004-8239-9410
Yu Zhang (iD) https://orcid.org/0000-0002-4888-5923

### Ethics

All animal experiments were conducted following the Ministry of Health national guidelines for housing and care of laboratory animals and performed in accordance with institutional regulations after review and approval by the Institutional Animal Care and Use Committee at the National Institute of Biological Sciences and Chinese Institutes for Medical Research. The assigned approval/accreditation number: AEEI-2023-223.

Reviewer #1 (Public review): https://doi.org/10.7554/eLife.108724.3.sa1
Reviewer #2 (Public review): https://doi.org/10.7554/eLife.108724.3.sa2
Author response https://doi.org/10.7554/eLife.108724.3.sa3

## Additional files

### Supplementary files

Supplementary file 1. List of sequences of gRNAs and primers used in current study.

Supplementary file 2. List of plasmids used in current study.

Supplementary file 3. Information of custom library used in current study.

Supplementary file 4. Results of ex vivo CRISPR/Cas9 genome-wide screenings in *Figure 1b*.

Supplementary file 5. Results of in vivo CRISPR/Cas9 screenings with custom library in *Figure 2c*.

MDAR checklist

### Data availability

The raw sequence data reported in the current study have been deposited in the Genome Sequence Archive in BIG Data Center, Beijing Institute of Genomics (BIG), Chinese Academy of Sciences, under accession numbers PRJCA010494 that can be accessed at https://ngdc.cncb.ac.cn/gsa/search?searchTerm=CRA007622. Other materials used during the current study are available from the corresponding author on reasonable request.

The following dataset was generated:

| Author(s) | Year | Dataset title | Dataset URL | Database and Identifier |
|---|---|---|---|---|
| Zhang Y | 2026 | OT-1 T cell RNA-seq data | https://ngdc.cncb.ac.cn/gsa/search?searchTerm=CRA007622 | National Genomics Data Center, CRA007622 |

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
