## [Editor Report · eLife Assessment]

The **important** study uses genome-wide CRISPR/Cas9 screenings to identify a novel target B4GALTZ1 that is implicated in modulating CD8+ T cell function in the context of anti-tumor immunity. The strength of evidence is **solid** but could benefit from more detail, particularly to verify the efficiency of knockout in their single gene KO lines and identification of N-glycosylation sites of TCR and CD8s. This work highlights the role of protein N-glycosylation, particularly B4GALT1 deficiency, in regulating CD8 function and anti-tumor immunity.

---

## [Referee Report · Reviewer #1 (Public review)]

Summary:

The study by Yu et al investigated the role of protein N-glycosylation in regulating T-cell activation and functions is an interesting work. By using genome-wide CRISPR/Cas9 screenings, authors found that B4GALT1 deficiency could activate expression of PD-1 and enhance functions of CD8+ T cells both in vitro and in vivo, suggesting the important roles of protein N-glycosylation in regulating functions of CD8+ T cells, which indicates that B4GALT1 is a potential target for tumor immunotherapy.

Strengths:

The strengths of this study are the findings of novel function of B4GALT1 deficiency in CD8 T cells.

Weaknesses:

Although authors have partly addressed my questions, including potential mechanism, however, I found that the impact of B4GALT1 deficiency for T cell function against tumor cells was not very striking, in comparing to other recently identified genes, which may limit its application, such as in adoptive T cell therapy.

Comments on revisions:

Authors have addressed the questions raised in previous review.

---

## [Referee Report · Reviewer #2 (Public review)]

Summary:

In this study, the authors identify the N-glycosylation factor B4GALT1 as an important regulator of CD8 T-cell function.

Strengths:

The use of complementary ex vivo and in vivo CRISPR screens is commendable and provides a useful dataset for future studies of CD8 T-cell biology.

The authors perform multiple untargeted analyses (RNAseq, glycoproteomics) to hone their model on how B4GALT1 functions in CD8 T-cell activation, as well as the use of a CD8-CD3 to narrow down the effects of B4GALT1, which is a broad-acting enzyme.

B4GALT1 is shown to be important in both in vitro T-cell killing assays and a mouse model of tumor control, reinforcing the authors' claims.

Weaknesses:

The authors did not verify the efficiency of knockout in their single gene KO lines, although they mention a plan to include such data in a future version of the manuscript.

The specific N-glycosylation sites of TCR and CD8 are not identified, and would be helpful for site-specific mutational analysis to further the authors' model.

The study or future studies could benefit from further in vivo experiments testing the role of B4GALT1 other physiological contexts relevant to CD8 T cells, for example autoimmune disease or infectious disease.

Comments on revisions:

The paper improved after revision.

---

## [Author Response]

The following is the authors’ response to the original reviews.

**eLife Assessment**
This valuable work investigates the role of protein N-glycosylation in regulating T-cell activation and function and suggests that B4GALT1 is a potential target for tumor immunotherapy. The strength of evidence is solid, and further mechanistic validation could be provided.

We sincerely thank the editor and reviewers for their time and constructive feedback. Your recognition of our work is much appreciated. We clarify our mechanistic studies as stated below.

**Public Reviews:**

**Reviewer #1 (Public review):**
Summary:The study by Yu et al investigated the role of protein N-glycosylation in regulating T-cell activation and functions is an interesting work. By using genome-wide CRISPR/Cas9 screenings, the authors found that B4GALT1 deficiency could activate expression of PD-1 and enhance functions of CD8+ T cells both in vitro and in vivo, suggesting the important roles of protein N-glycosylation in regulating functions of CD8+ T cells, which indicates that B4GALT1 is a potential target for tumor immunotherapy.Strengths:The strengths of this study are the findings of novel function of B4GALT1 deficiency in CD8 T cells.Weaknesses:However, authors did not directly demonstrate that B4GALT1 deficiency regulates the interaction between TCR and CD8, as well as functional outcomes of this interaction, such as TCR signaling enhancements.

We are very sorry that we did not highlight our results in Fig. 5f-h enough. In those figures, we demonstrated the interaction between TCR and CD8 increased significantly in B4GALT1 deficient T-cells, by FRET assays. To confirm the important role of TCR-CD8 interaction in mediating the functions of B4GALT1 in regulating T-cell functions, such as in vitro killing of target cells, we artificially tethered TCR and CD8 by a CD8β-CD3ε fusion protein and tested its functions in both WT and B4GALT1 knockout CD8^+^ T-cell. Our results demonstrate that such fusion protein could bypass the effect of B4GALT1 knockout in CD8^+^ T-cells (Fig. 5g-h). Together with the results that B4GALT1 directly regulates the galactosylation of TCR and CD8, those results strongly support the model that B4GALT1 modulates T-cell functions mainly by galactosylations of TCR and CD8 that interfere their interaction.

**Reviewer #2 (Public review):**
Summary:In this study, the authors identify the N-glycosylation factor B4GALT1 as an important regulator of CD8 T-cell function.Strengths:(1) The use of complementary ex vivo and in vivo CRISPR screens is commendable and provides a useful dataset for future studies of CD8 T-cell biology.(2) The authors perform multiple untargeted analyses (RNAseq, glycoproteomics) to hone their model on how B4GALT1 functions in CD8 T-cell activation.(3) B4GALT1 is shown to be important in both in vitro T-cell killing assays and a mouse model of tumor control, reinforcing the authors' claims.Weaknesses:(1) The authors did not verify the efficiency of knockout in their single-gene KO lines.

Thank reviewer for reminding. We verified the efficiency of some gRNAs by T7E1 assay. We will add those data in supplementary results in revised version later.

(2) As B4GALT1 is a general N-glycosylation factor, the phenotypes the authors observe could formally be attributable to indirect effects on glycosylation of other proteins.

Please see response to reviewer #1.

(3) The specific N-glycosylation sites of TCR and CD8 are not identified, and would be helpful for site-specific mutational analysis to further the authors' model.

Thank reviewer for suggestion! Unfortunately, there are multiple-sites of TCR and CD8 involved in N-glycosylation (https://glycosmos.org/glycomeatlas). We worry that mutations of all these sites may not only affect glycosylation of TCR and CD8 but also other essential functions of those proteins.

(4) The study could benefit from further in vivo experiments testing the role of B4GALT1 in other physiological contexts relevant to CD8 T cells, for example, autoimmune disease or infectious disease.

Thank reviewer for this great suggestion to expand the roles of B4GALT1 in autoimmune and infection diseases. However, since in current manuscript we are mainly focusing on tumor immunology, we think we should leave these studies for future works.

**Recommendations for the authors:**

**Reviewer #1 (Recommendations for the authors):**
The study by Yu et al investigated the role of protein N-glycosylation in regulating T-cell activation and functions is an interesting work. By using genome-wide CRISPR/Cas9 screenings, the authors found that B4GALT1 deficiency could activate expression of PD-1 and enhance functions of CD8+ T cells both in vitro and in vivo, suggesting the important roles of protein N-glycosylation in regulating functions of CD8+ T cells, which indicates that B4GALT1 is a potential target for tumor immunotherapy. However, authors need to directly demonstrate that B4GALT1 deficiency regulates the interaction between TCR and CD8, as well as functional outcomes of this interaction, such as TCR signaling enhancements. In addition, blocking PD1 has been shown to enhance antitumor effect, whereas the presented data in this study suggest that the activation of PD1 expression in the condition of B4GALT1 deficiency in T cells enhanced antitumor effect. How to reconcile this discrepancy? Finally, several minor questions need to be addressed to strengthen the conclusions in this manuscript.

(1) We used a FRET (Fluorescence Resonance Energy Transfer) assay to measure interaction between TCR and CD8. FRET signals of TCR-CD8 increased significantly in B4GALT1 deficient T-cells, compared with control cells (Fig. 5f). For functional outcomes of this interaction, we observed enhanced T-cell killing activities in B4GALT1 deficient CD8^+^ T-cells (Fig. 3f and Fig. 5h).

To confirm whether reduced TCR-CD8 interaction is the major cause of TCR activation phenotypes in B4GALT1 knockout CD8^+^ T-cells, we generated a construct in which we fused the CD8b ectodomain (ECD) with CD3e to artificially tether TCR with CD8 (Fig.5g). Overexpression of such CD8β-CD3ε fusion led to enhanced in vitro killing activities in control wild-type CD8^+^ T-cells. On the other hand, in B4GALT1 deficient CD8^+^T-cells, such enhanced T-cell killing activities by fusion construct was significantly diminished (Fig.5h), suggesting it bypassed the regulation by B4GALT1.

(2) PD-1 is both an early T-cell activation marker upon TCR activation and a T-exhausted marker under consecutive or repeated stimulations. In our screenings, PD-1 was used as an early activation marker for T-cells.

We have clarified this in new Discussion section.

(1) The present data relies on statistical graphs (e.g., bar and line charts) for all data, excluding the bioinformatics analysis. Including data such as flow cytometry plots, photomicrographs, or immunohistochemistry staining images will provide more direct support for the conclusions.

Thank the reviewer for valuable suggestions! We added original flow cytometry gating strategies for Cas9 screening sorting (Fig. S1a), TIL analysis (Fig.S5), and FRET assay (Fig. S8) in revised version to provide more direct support for our conclusions.

(2) To further validate the enhanced tumor infiltration phenotype resulting from B4GALT1 knockout, the following data would strengthen the manuscript:(a) Flow cytometric analysis of TILs or immunofluorescence data from tumor sections.

Thank the reviewer for valuable suggestion! We added original flow cytometry gating strategies for TILs in Fig. S5 in revised version.

(b) Assessment of in vivo T cell proliferation, for example, by tracking changes in the proportion of CD8+ T cells in the peripheral blood over time.

We analyzed in vivo T-cell proliferation within tumor by CFSE (carboxyfluorescein succinimidyl ester) analysis. As shown in Fig. S6b, 6 days after infusion, B4GALT1 knockout OT-I T-cell showed increased proliferation within tumors, comparing with wild type control OT-I cells.

(c) Evaluation of the proliferation and activation status of OT-1 CD8+ T cells specifically in the draining lymph nodes of the mouse model.

Thank the reviewer for valuable suggestion! We plan to perform this experiment in the future.

(3) The authors provide evidence that B4GALT1 knockout enhances CD8+ T cell function in both mouse models and human TCR-T cells (in vitro). Definitive support for the translational potential of this strategy would come from showing that B4GALT1-knockout human TCR-T cells also mediate potent in vivo function (NSG tumor-bearing model may be a better choice).

Thank the reviewer for valuable suggestion! We are going to perform those experiments in the future. However, we do not expect that in vitro and in vivo (NSG mice) experiments will show much different results, which may also not add too much for current manuscript.

(4) It would be preferable to include data on T cell activation and effector function (e.g., flow cytometry for IL-2, TNF-α, and IFN-γ, or ELISPOT) following stimulation with an OVA-specific peptide or co-culturing of OVA-expressing tumor cells with B4GALT1-knockout OT-1 CD8 T cells, especially the changes in the TILs compared with the non-targeting control group.

Following co-culturing of B16-OVA tumor cells with B4GALT1-knockout or wild-type OT-I CD8^+^ T-cells, the RNA levels and secretion levels of TNFα and IFNγ were detected by RT-qPCR and ELISA, respectively (Fig. 3c). B4GALT1-deficient OT-I T-cells showed increased expression of T-cell activation and cytotoxic markers such as IFNγ and TNFα.

(5) What is the correlation between the expression of B4GALT1, PD-1, and TCR activation markers at various time points during a long-term T cell co-culture with tumor cells?

Thanks for the reviewer for valuable suggestion! We don’t have this data now. While we agree that exploring this might be interesting, we think it falls outside the scope of the current study.

(6) In line 136: Regarding the genetic targeting of B4GALT1 in T cells, it is unclear whether single or multiple gRNAs were used and if potential off-target effects were assessed. To fully validate the model, it would be important to clarify these strategies, and it is essential to include data on the knockout efficiency at both the protein (e.g., Western blot) and mRNA levels.

We are sorry about the unclear statements for gene knockout strategy. In current study, single sgRNAs were used in all experiments for gene knockout. B4galt1 sg2 was used in Fig. 3a. Both B4galt1 sg1 and sg2 were used in Fig. S1d. We clarified this in each figure legend in revised version.

The phenotypes of B4galt1 knockout T-cells could be rescued by overexpression of either a short or long isoform of mouse B4galt1 cDNA (Fig. 3b), indicating that potential off-target effects could be excluded.

The sgRNA knockout efficiencies were confirmed by T7E1 assay in revised version (Fig. S2). Regrettably, anti-mouse B4galt1 antibody didn’t work in western blot.